# Additive Manufacturing of Electrically Conductive Multi-Layered Nanocopper in an Air Environment

**DOI:** 10.3390/nano14090753

**Published:** 2024-04-25

**Authors:** David Pervan, Anil Bastola, Robyn Worsley, Ricky Wildman, Richard Hague, Edward Lester, Christopher Tuck

**Affiliations:** 1Centre for Additive Manufacturing, Faculty of Engineering, University of Nottingham, Nottingham NG7 2RD, UK; david.pervan@nottingham.ac.uk (D.P.); anil.bastola@nottingham.ac.uk (A.B.); robyn.worsley@nottingham.ac.uk (R.W.); ricky.wildman@nottingham.ac.uk (R.W.); richard.hague@nottingham.ac.uk (R.H.); 2Advanced Materials Research Group, Faculty of Engineering, University of Nottingham, Nottingham NG7 2RD, UK; edward.lester@nottingham.ac.uk

**Keywords:** additive manufacturing, inkjet, copper, nanoparticles, multi-layer

## Abstract

The additive manufacturing (AM) of functional copper (Cu) parts is a major goal for many industries, from aerospace to automotive to electronics, because Cu has a high thermal and electrical conductivity as well as being ~10× cheaper than silver. Previous studies on AM of Cu have concentrated mainly on high-energy manufacturing processes such as Laser Powder Bed Fusion, Electron Beam Melting, and Binder Jetting. These processes all require high-temperature heat treatment in an oxygen-free environment. This paper shows an AM route to multi-layered microparts from novel nanoparticle (NP) Cu feedstocks, performed in an air environment, employing a low-power (<10 W) laser sintering process. Cu NP ink was deposited using two mechanisms, inkjet printing, and bar coating, followed by low-power laser exposure to induce particle consolidation. Initial parts were manufactured to a height of approximately 100 µm, which was achieved by multi-layer printing of 15 (bar-coated) to 300 (inkjetted) layers. There was no evidence of oxidised copper in the sintered material, but they were found to be low-density, porous structures. Nonetheless, electrical resistivity of ~28 × 10^−8^ Ω m was achieved. Overall, the aim of this study is to offer foundational knowledge for upscaling the process to additively manufacture Cu 3D parts of significant size via sequential nanometal ink deposition and low-power laser processing.

## 1. Introduction

Additive manufacturing (AM) or 3D printing is a process whereby objects are created in a layer-by-layer fashion directly from computer-aided designs (CAD). Several different AM techniques are available to manufacture functional parts from metals, polymers, polymer composites, and ceramic materials. According to ASTM standards, there are seven AM techniques, including Binder Jetting, Electrochemical Deposition, Direct Energy Deposition, Material Extrusion, Material Jetting, Powder Bed Fusion, Sheet Lamination and Vat Photopolymerization [1,2,3,4,5,6,7,8,9,10,11]. In recent years, there has been significant interest in the use of AM across many fields due to its advantages over conventional subtractive manufacturing. There is no need for tooling, which allows ‘freedom of design’, high customisation levels, the potential for less wastage, and a significantly reduced ‘time to market’. 

The potential for AM of functional copper (Cu) parts has gained considerable interest, particularly in electronic devices, due to the high conductivity (comparable to silver) as well as the low cost and high abundance of the material [12,13,14,15]. However, one of the biggest challenges with Cu is that it can be easily oxidised during the sintering process. Cu can easily oxidise at room temperature and become insulating, which is undesirable for the maintenance of high electrical conductivity. Oxidation can be prevented by reducing the oxygen concentration during the sintering process using a constant inert gas flow (i.e., argon or nitrogen) or by using a vacuum. However, this approach increases the overall complexity and cost of the process. Low-power sintering (<10 W) with a laser system could potentially work in air (and avoid oxidation) and would certainly present a lower-cost alternative. The lower-power laser potentially reduces the thermal stress build-up and may reduce the need for support structures in complex ‘builds’ and the machinery’s cost and complexity.

There are several reported studies that develop Cu functional parts using different AM techniques. However, previous work on AM of multi-layered Cu parts has concentrated on high-energy manufacturing processes such as SLM [16,17,18] and EBM [19,20,21]. Cu parts have also been produced using binder jetting, which is a relatively low-energy process. However, it does require high-temperature post-printing heat treatment [22]. All of the above-mentioned AM techniques describe the sintering of micron-sized Cu particles to obtain the final components. 

Nanoparticles (NP), on the other hand, generally require much lower sintering temperatures than micron-sized particles. Some studies dedicated to NP sintering using ultra-fast photonic treatments have demonstrated that the Cu NP can be sintered in an air environment without oxidation [23,24,25,26]. Moreover, Kim et al. [27] successfully reported the oxidation-free sintering of Cu NP in an ambient environment but concluded that a sintering time in the millisecond range was required to avoid oxidation.

Inkjet printing is a widely used process that underpins a number of AM techniques and has significant relevance to the processing of nanoparticulate-based inks, where inks are deposited onto a substrate through an array of micron-sized nozzles [15,28,29,30]. Inkjet printing can achieve high resolution with customized inks [1,31]. Similar to other composite inks, including Ag-based inks [32,33] and graphene-based inks [34,35,36], Cu NP inks can be formulated by mixing the NP with suitable solvents for the inkjet printing process [17,37,38]. Notably, inkjet printing has been successfully used to print up to 1000 layers of Ag NP ink, which were subsequently sintered to create multi-layered functional Ag parts [39,40]. Similarly, Cu NP inks have also been printed and sintered using inkjet printing [24], but with a maximum of only four print layers. In a more recent study, inkjet printing of up to 10 layers of Cu NP was demonstrated [14], with thermal sintering used to consolidate the particles.

The main aim of this study was to print multiple layers of Cu via inkjet printing and successfully sinter the Cu NP using low laser powers in an air environment to create multi-layered Cu micro parts (Figure 1). A bespoke sintering setup was developed to induce particle consolidation after printing. Cu samples were produced using inkjet printing and then compared to those fabricated via a bar coating technique. Thereafter, various properties of the sintered multi-layered Cu materials, developed via both inkjet printing and bar coating, were characterised using a range of techniques. Oxidation, morphology, indentation hardness, surface roughness, and electrical properties of the printed and sintered Cu materials are reported, relative to the values for bulk Cu.

## 2. Materials and Methods

### 2.1. Materials 

Two types of Cu nanoparticle suspension (35–37 wt.% metal loading) were obtained from a commercial ink supplier in Nottingham, UK. The solvent in the suspension used for inkjet printing was 2-(2-ethoxyethoxy) ethanol (C_6_H_14_O_3_), also known as diethylene glycol mono-ethyl ether (DGME). The solvent in the suspension used for bar coating was propan-2-ol (C_3_H_8_O) with the addition of 5 g L^−1^ 2-hydroxypropanoic acid. Both inks had a viscosity of 22 cP at 25 °C, a surface tension of 35.2 mN m^−1^, and were loaded with Cu NP with an average particle size of 330 (±186) nm according to the supplier. 

### 2.2. Inkjet Printing

The 1 mm^2^ rectangles were inkjet-printed using a commercially available Dimatix inkjet printer (DMP-2900 Series, Fujifilm Dimatix, Santa Clara, CA, USA). Drop spacing was set to 40 µm in order to produce uniform print lines. After each printed layer, the ink was left to dry on the glass substrate for 90 s at 60 °C.

### 2.3. Bar Coating

A 20 µm metal ‘K’ bar from RK PrintCoat Instruments Ltd. (Lillington, UK) was used for bar coating. The glass substrate was firmly clamped, and the bar-coated formulation of the Cu ink was dispensed onto the substrate. The hand printer was then manually pulled backwards to spread the ink across the substrate. After the bar coating, the deposited ink was left to dry for 30 s. Each dry deposited ink layer was found to be ~7 µm thick.

### 2.4. Thermal Characterisation

Cyclic Differential Scanning Calorimetry (DSC) measurements were carried out to obtain the sintering temperature range. The sample ink was placed into an alumina DSC pan and dried in an oven at 40 °C to evaporate residual solvents. A 5 mg sample was heated in a TA Q600 (TA Instruments, New Castle, DE, USA) from room temperature, increasing to 1200 °C, before cooling to 150 °C and subsequently increasing back to 1200 °C at 10 °C min^−1^, all in a nitrogen environment.

### 2.5. Lower Power Laser Sintering 

A custom-built low-power laser sintering system was used to selectively sinter the Cu NP. The system consisted of a fibre laser (BKtel GmbH, Hückelhoven, Germany) with ≤10 W power at a wavelength of 1064 nm, a scan head (hurrySCAN, Scanlab GmbH, Puchheim, Germany), a 164 mm T-theta lens, and control software (laserDESK v.1, Scanlab GmbH, Puchheim, Germany). The laser power used was 2.5 W with a scan rate of 150 mm s^−1^. The hatch distance between the parallel laser scan lines was set to 50 µm. Laser irradiation to sinter Cu NP was performed in ambient conditions. See Appendix A for details of the bespoke system for the deposition and sintering of Cu NP.

### 2.6. Particulate and Surface Characterisation

Various techniques were used to characterise the pre- and post-sintered samples. 

XRD analysis was carried out to study oxidation and average particle size, pre- and post-sintering, using a Bruker D8 Advance instrument (Bruker Corp., Billerica, MA, USA). 

Nanoindentation was used to measure the structural behaviour of the Cu samples using a NanoTest P3 (Micro Materials Ltd., Wrexham, UK).

White Light Interferometry (WLI) measurements were carried out to quantify and compare the surface roughness of Cu samples using an Infinite Focus G5 microscope (Bruker Alicona, Bruker Corp., Billerica, MA, USA).

Cross-sections of the Cu samples were imaged using a JEOL 7100F (JEOL Ltd., Tokyo, Japan) Scanning Electron Microscope (SEM), and ImageJ was used to quantitatively measure sample porosity. 

Sheet resistance measurements were carried out using a four-point probe (Measuring Device Type SD-800, Measuring Probe Type: SDKR-13, NAGY Messsysteme GmbH, Gäufelden, Germany) to measure the sheet resistance of the sintered Cu samples. 

All experimental tests were carried out in triplicate and presented as averaged data. 

## 3. Results and Discussion

### 3.1. NP Sintering

Metal particle sintering is an irreversible exothermic process during which particles release surface energy to build bonds between particles [41]. Cu NP sintering is identifiable by the exothermic portion of the DSC measurement peak, shown in Figure 2, between approximately 155 and 597 °C This peak only occurred in the first DSC temperature ramp and did not occur in the second and third temperature ramps, meaning that the melting process is irreversible. The small shoulder prior to 155 °C is believed to be the result of residual solvent being driven off. 

SEM images of un-sintered and laser-sintered Cu NP are given in Figure 3a and 3b, respectively. The images show a relatively wide particle range, with most particles being sub-microns (scale bar 1 micron). The NP fuse together during the laser sintering process, resulting in increased particle size and decreased spatial homogeneity. The XRD patterns in Figure 3c do not show any signs of oxidation, even after sintering. Copper oxide characteristic peaks should appear at 2θ values of 32.5°, 35.5°, 38.7°, 48.6°, 53.5°, 58.2°, 61.4°, 65.8°, 67.8° [42]. The Scherrer Equation was used to calculate particle size (Appendix A). The laser sintering produced (on average) a 37% increase in mean Cu crystallite size (Figure 3d). 

### 3.2. Properties of Printed Cu

#### 3.2.1. Morphology

Figure 4 shows a collection of SEM images for both inkjet-printed (Figure 4a,c,e) and bar-coated (Figure 4b,d,f) Cu materials. It is apparent from these images that the top surface of the sample produced by inkjet printing is uneven (Figure 4a). Jetting onto previous layers, which were not entirely dry, potentially results in the migration of particles towards the edges of wet areas. This effect is a consequence of capillary pressure and is a phenomenon occurring during single droplet deposition in inkjet printing, known as the ‘coffee ring effect’ [1]. However, these samples will also be subject to Marangoni currents created within a drying liquid during multi-layer deposition due to convection currents.

Figure 4c shows a higher magnification, where the internal microstructure of the inkjet-printed Cu is inhomogeneous. Particle sizes appear to be predominantly in the nano-range but with a few particles significantly larger (at the micron scale) and randomly scattered within the cross-section. Larger pores (50 µm), mostly horizontally orientated, can be seen across the whole cross-section. In contrast, the bar-coated Cu samples (Figure 4d) displayed relatively uniform porous internal micro-structuring, characterised by particles and pores on a scale ranging from nanometres to a few microns. 

ImageJ V1.52 was used to quantify the internal porosity of the SEM cross-sections (Figure 4c,d) for the multi-layered printed Cu materials, with the results given in Figure 5a. The results show a very similar average porosity (~63%) for both the bar-coated and the inkjet-printed samples. However, the standard deviation associated with the porosity measurements of the inkjet-printed samples is significantly greater than the bar-coated samples, indicating that the difference in internal porosity is statistically significant between the two methods. It is hypothesised that the inkjet-printed layers were not completely dry when they were sintered. Laser irradiation then caused the residual solvent to rapidly volatilise, creating an inhomogeneous internal structure and variability of porosity in the sample. Moreover, the drying mechanisms (within the inkjet process) may also be a cause for the higher porosity, as the nanoparticles migrate towards the edges of the droplets, inducing inhomogeneous internal structures (Figure 4c).

#### 3.2.2. Surface Roughness

The thickness of the final samples, both inkjet and bar-coated, was approximately 100 µm. For each sample, 15 and 300 layers were required on average to reach the desired height of 100 µm when using the bar coating and inkjet inks, respectively. Figure 5b shows the surface roughness (mean height) of Cu parts. The inkjet-printed samples were more heterogeneous than the bar-coated samples (on average, the roughness was 53.5% greater). This might also be a result of the accumulated roughness over the significantly larger number of inkjetted layers required to print a 100 µm-thick sample, i.e., the inkjetted layers were, on average, 20 times thinner than the bar-coated layers. The linear relationship between surface roughness and number of layers also applies to bar-coated samples. This linear relationship suggests that newly added layers do not “repair” the previous layer but rather that each layer cumulatively adds surface roughness. It was found that the bar-coated and inkjet-printed samples have an average surface roughness per layer of 1.3 µm and 0.1 µm, respectively.

#### 3.2.3. Nanoindentation

Mechanical part properties are important for applications for potential upscaling of the process to significant 3D part heights. For both bar-coated and inkjet-printed samples, indentation hardness and creep were investigated and presented as a fraction of the bulk Cu properties (Figure 5c,d). See Appendix A for the details of hardness and creep measurements. The predominant deformation mechanisms of the bar-coated and inkjet-printed samples appeared to be viscous, elastic, and plastic deformations [43]. Figure 5c shows that the indentation hardness of the bar-coated samples was, on average, 55.8% higher than the indentation hardness of the inkjet-printed samples. 

The standard deviation in the indentation hardness of the Inkjet-printed sample was significantly higher than the standard deviation of the bar-coated sample (400%), which suggests a strong non-uniformity in the inkjet-printed samples, similar to that seen with surface roughness measurements. The average maximum indentation depth for the bar-coated and inkjet-printed samples was 16.4 µm and 13.6 µm, respectively, while the arithmetical mean roughness for the bar-coated and inkjet-printed samples was 19.0 µm and 29.2 µm, respectively. This signifies that the average surface roughness is even greater than the average maximum indentation depths. However, it should be noted that the indentation hardness in both samples was just below 1% of the measured indentation hardness of bulk Cu samples. Such low hardness is attributed to the high porosity of the samples [44]. Elsewhere, Jiang et al. [45] also studied the effect of surface roughness on the indentation hardness of Cu and also demonstrated, through simulations and experiments, that the increased surface roughness considerably reduced indentation hardness.

#### 3.2.4. Electrical Properties

A four-point probe was used to carry out sheet resistance measurements for the multi-layered printed samples as a function of the build height (Figure 6a). By taking into account the spatial dimension of the samples, the sheet resistance measurements were also converted to resistivity (Figure 6b).

Sheet resistance decreases with increasing build height for both bar-coated and inkjet-printed samples. The sheet resistance of the bar-coated sample was approximately one order of magnitude lower across the entire build height when compared to the inkjet-printed sample. 

Resistivity was also plotted as a multiple of bulk Cu resistivity (1.68 × 10^−8^ Ω m), which describes the resistance for a material of unit length and unit cross-sectional area. Figure 6b shows that the resistivity increases with increasing build height for both bar-coated and inkjet-printed samples. However, the samples (produced using the two different methods) were fabricated to achieve a specific total thickness, i.e., 100 µm, with approximately the same weight loading in the ink. The nature of deposition was quite different, with a higher degree of ‘packing’ resulting from the bar coating method, which may have potentially created more particle/particle contact and a higher densification of the particles in the layer. Even though 100 µm thickness was achieved in both cases, the ink-jetted copper required 20× as many layers. The resistivities across the samples (as the thickness of the samples was increasingly stacked up) were, on average, 11.7 times higher for the inkjet-printed samples compared to the bar-coated samples. Furthermore, the relative increase in resistivity across the build height of the inkjet-printed samples was greater than the equivalent for the bar-coated samples. The cross-sectional SEM images in Figure 4 show the interlayers to be less dense for inkjet-printed samples. It is believed that the interlayers resisted the flow of electricity more than the intra-layers due to the increased porosity. The inkjet-printed material also contained a greater fraction of interlayers with respect to vertical height due to the increased number of layers per unit height. This may explain why there was a greater increase in resistivity across the build height of the inkjet-printed samples compared with the bar-coated samples. 

Previous work using ink-jetted Ag NP parts (1000 layers) used longer sintering to produce resistivity values of only 8 to 14 times higher than bulk Ag [39]. The bar coating and laser sintering process was able to produce multi-layered structures with resistivities in the same range (~17 times higher than that of bulk Cu), in agreement with previous studies. However, it should be noted that we were able to achieve this resistivity with sintering times approximately 4 orders of magnitude faster.

### 3.3. Sintering and Particle Size

The temperature required to induce sintering is highly dependent on particle size [46]. The average Cu NP size employed in this study was ~330 nm (Figure 3a). DSC measurements (Figure 2) show that temperatures must be above approximately 155 °C to start sintering the nanoparticles in the deposited layer(s). These melt layers should also be cooled down as quickly as possible (ideally in milliseconds) to avoid oxidation. The energy sources widely reported to be capable of inducing sufficiently high heating and cooling rates are photonic sources, in particular lasers and intense pulsed flashlights (IPL) [23,24]. 

Previous reports on additively manufactured electrically conductive multi-layered metal parts report significantly longer sintering times than those employed within this work. This is because they either use a different metal (which does not readily oxidise) or the process is carried out in a low-oxygen environment. The most directly comparable studies by Vaithilingam et al. [39,40] report inkjet printing and sintering of up to 1000 layers of Ag NP ink. In this case, an IR lamp was utilised for sintering, and the reported IR system (1000 W power) was non-selective, with an exposure time of more than 1 s. In other cases, AM of conductive multi-layered metal parts is achieved using SLM techniques, which commonly use powders with particle sizes in the low micron range (30 to 80 µm) [47] and generally require considerably higher laser powers, e.g., Singer et al. [48] reported the use of laser powers of 1000 W.

In this study, we found that a laser power of 2.5 W with an in-focus spot size diameter of 32 µm was sufficient to induce sintering. The average exposure time was approximately 0.17 ms. Two independent variables were investigated to optimise the laser sintering process: (1) laser power and (2) laser speed. Figure 7 provides contour plots of these two process parameters analysed in terms of the uniformity of sintered lines. In the optimization process, single laser line scans were carried out. Four different types of the sintered line were identified under the microscope, which were differentiated as evenly sintered uniform lines (light blue colour), strongly sintered or melted lines (green colour), continuously ablated and melted lines (yellow colour), and ablated lines (orange colour). It is noted that increasing laser irradiance and decreasing laser speed increases the energy exposure. For an increased film thickness, the energy delivered by the laser needs to cover a larger volume of copper. Essentially, the energy density per unit volume decreases with increasing film thickness. 

The relationship between the laser energy input and the width of the sintered area was investigated. For all the laser sintering carried out for the process window maps, the energy input was plotted against the sintered line width. The laser fluence was calculated using Equation (1):(1)$$F=\frac{P}{U\;D}$$
where F is the laser fluence (J cm^−2^), P is the laser power (W), U is the laser velocity (mm s^−1^), and D is the laser spot diameter (on average 32 µm).

An optical microscope (Nikon Eclipse LV100ND, Melville, NY, USA) was used to take images of each sintered line to measure the sintered line width. The width of the sintered lines was then averaged on a pixel basis using MATLAB 9.6 image analysis. The narrowest sintered (and conductive) line measured was approximately 15 µm wide. After a thorough investigation into the relationship between laser energy input and the consequences for sintered lines, represented in Figure 8, it was found that a laser fluence of up to 7 J cm^−2^ can effectively produce uniform and evenly sintered lines. This is equivalent to a laser power as low as 1 W at a laser speed of 450 mm^−1^ (marked by the dotted blue line in Figure 8) and sufficient to induce Cu NP’s sintering into wide lines. As mentioned earlier, crystal size (Figure 3b) increased by 37% as a result of laser sintering. To date, there is no literature quantifying the crystal size increase as a result of laser sintering.

## 4. Conclusions

A novel low-power (2.5 W) sintering process has been successfully demonstrated to achieve the additive manufacture of electrically conductive multi-layered Cu materials in an air environment.

Two different Cu ink deposition methods, inkjet printing, and bar coating, were employed and compared. The bar coating approach was more time efficient, as solvents with higher volatility can be used and deposited layers are significantly thicker than ink-jetted layers. 

The bar-coated Cu materials had better electrical properties and more robust mechanical properties than ink-jetted Cu materials. The sintered Cu materials, regardless of the deposition method, were found to be unoxidised but characterised as low-density and highly porous structures. 

Sintering Cu NP in an ambient environment necessitates very short sintering times to avoid oxidation. Shorter residence times limit the degree of densification. Clearly, the structures’ low density and porosity result in physical properties that are not close to those of bulk Cu. However, the electrical properties were not significantly influenced; electrical conductivities close to bulk Cu were still achieved. 

Potential future work could be focused on scaling up the process to further increase the number of layers as well as achieving increased densification through other techniques, such as heat treatment of the completed materials and better ink formulations. It is hoped that the knowledge created in this work will enable AM of multi-layered Cu materials, with sufficient properties close to those of bulk Cu, directly from NP feedstock in an air environment and at a significantly reduced cost compared to current AM processes.

## Figures and Tables

**Figure 1 nanomaterials-14-00753-f001:**
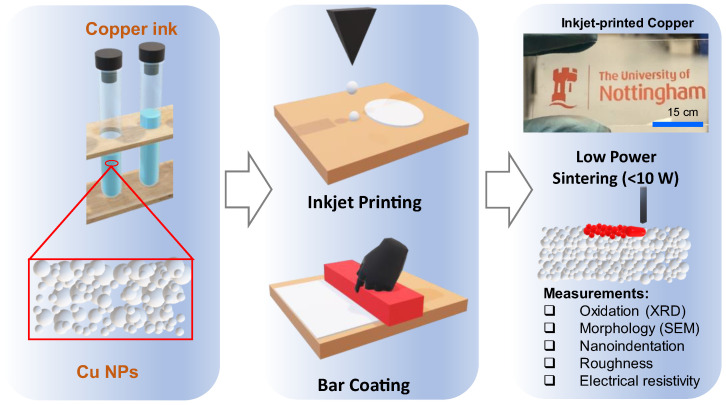
An overview of the investigation carried out in this study. Cu micro parts were developed through inkjet printing and bar coating via the direct deposition of Cu NP inks followed by low-power photonic sintering. Afterwards, various properties of the developed Cu microparts were investigated.

**Figure 2 nanomaterials-14-00753-f002:**
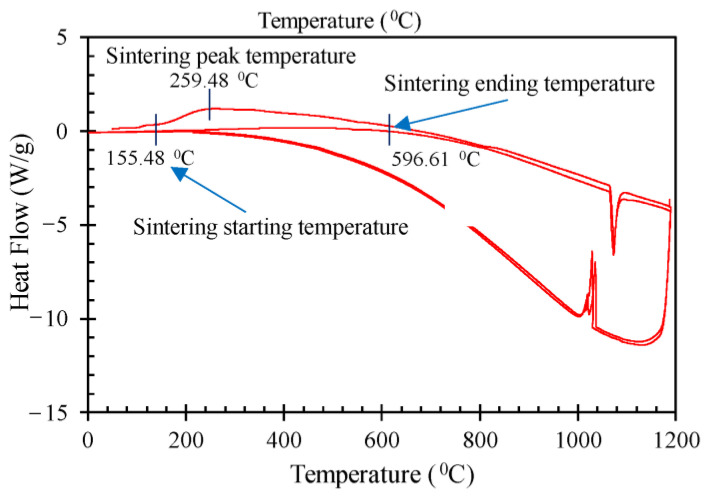
DSC results for Cu NP, weight corrected heat flow plotted against temperature.

**Figure 3 nanomaterials-14-00753-f003:**
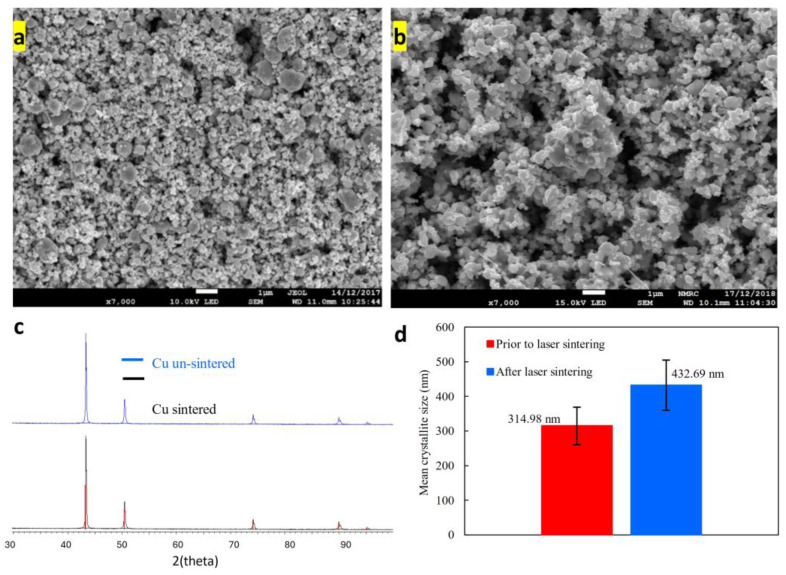
SEM images of un-sintered (**a**) and laser processed (**b**) Cu NP. The scale bar is 1 µm. (**c**) XRD patterns of sintered and un-sintered Cu. (**d**) Mean Cu NP crystallite size before and after laser sintering.

**Figure 4 nanomaterials-14-00753-f004:**
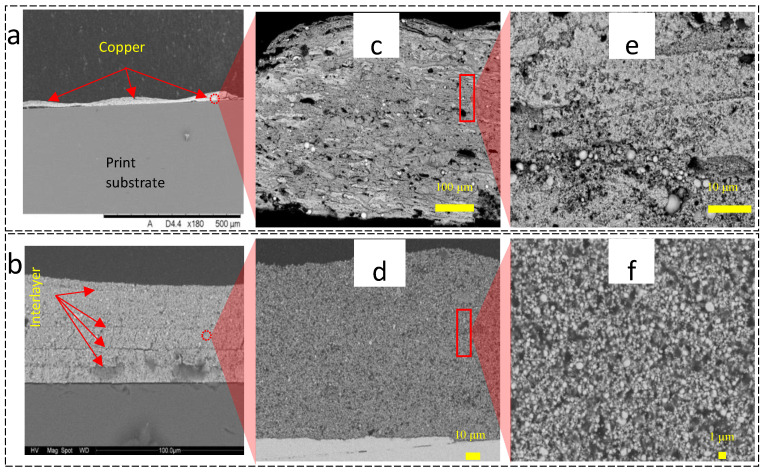
Compilation of SEM images of the cross-section of (**a**,**c**,**e**) inkjet-printed and (**b**,**d**,**f**) bar-coated Cu samples.

**Figure 5 nanomaterials-14-00753-f005:**
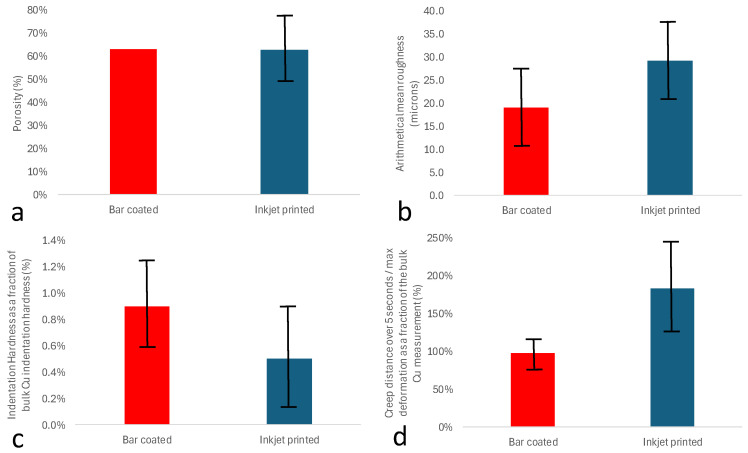
Properties of printed and sintered Cu samples. (**a**) Comparison of porosity for bar-coated and inkjet-printed samples obtained through cross-sectional SEM image analysis (Figure 4). (**b**) Arithmetical mean height (surface roughness) of the top surface of 100 µm thick bar-coated and inkjet-printed samples. (**c**) Indentation hardness. (**d**) Creep distance over 5 s as a fraction of maximum plastic deformation. The results are presented as a fraction of the bulk Cu measurement. The standard deviation for each test is plotted as an error bar.

**Figure 6 nanomaterials-14-00753-f006:**
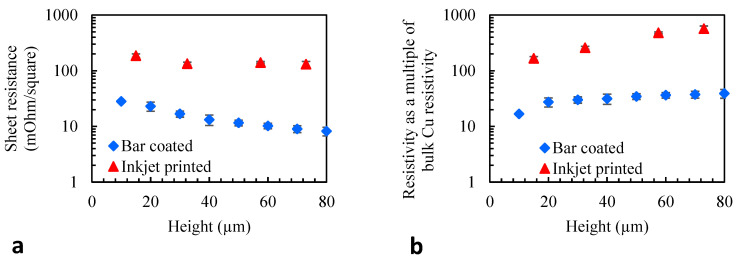
Results of electrical measurements. (**a**) Sheet resistance against build height and (**b**) resistivity (as a multiple of bulk Cu resistivity) against build height for both bar coating and inkjet printing. Note, the y-axis is on a logarithmic scale for better visualisation. Error bars indicate standard deviations.

**Figure 7 nanomaterials-14-00753-f007:**
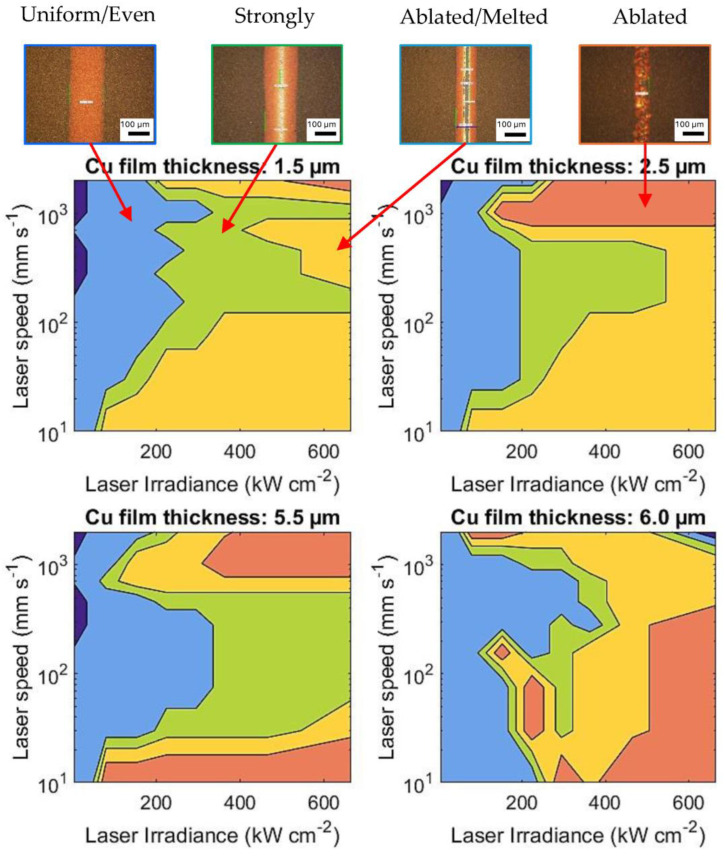
Process window contour maps of different sintering phenomena at varying laser irradiance and laser speeds for four different Cu film thicknesses on a glass substrate. The images and labels indicate dark blue: no effect, light blue: evenly sintered, green: strongly sintered, yellow: continuously ablated and melted, red: ablated.

**Figure 8 nanomaterials-14-00753-f008:**
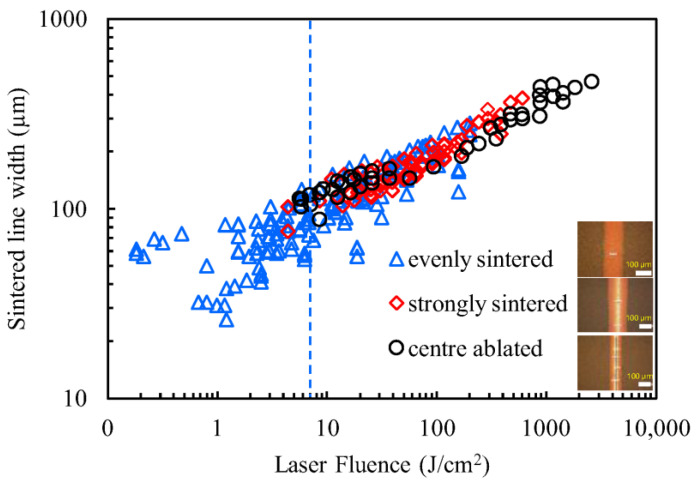
Sintered line width against laser fluence for a single layer of Cu NP.

## Data Availability

Data is contained within the article or Appendix A.

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
