# Peer review of "Additive Manufacturing of Electrically Conductive Multi-Layered Nanocopper in an Air Environment"

_nanomaterials, 2024, doi:10.3390/nano14090753_

Round 1
Reviewer 1 Report
Comments and Suggestions for Authors
Inkjet printing and bar coating methods combined with a low power laser sintering process are used for Cu nanoparticle feedstocks to fabricate functional copper parts. The microstructure features and resistivity property are studied for two methods, which can provides useful results for additively manufacture Cu 3D parts. Here are some weakness to be further clarified.
1. The text size of angles in abscissa axis of Fig. 3c are too small to be seen.
2. The specific difference in pore size between inkjet printing and bar coating is suggested to be provided.
3. It is stated that the thickness of the final samples, both inkjet and bar-coated, was approximately 100 µm. However, the thickness for inkjet printing seen from Fig. 4c is about 600 μm,please check the scale bar.
4. What is the relation between the high electrical resistivity and porous density/pore sizes? Also, what is the purpose of fabricating copper samples with high electrical resistivity?
5. The texts in Fig.5 are hard to been seen clearly.
6. In the conclusion, it is stated that “A novel low-power (1 W) sintering process has been successfully demonstrated…”. Whereas, the laser power used in this study is 2.5 W.
7. Please complete information of author contributions.
Author Response
REVIEWER 1
Inkjet printing and bar coating methods combined with a low power laser sintering process are used for Cu nanoparticle feedstocks to fabricate functional copper parts. The microstructure features and resistivity property are studied for two methods, which can provides useful results for additively manufacture Cu 3D parts. Here are some weakness to be further clarified.
- The text size of angles in abscissa axis of Fig. 3c are too small to be seen.
corrected
- The specific difference in pore size between inkjet printing and bar coating is suggested to be provided.
Pore size/porosity shown and quantified for sintered samples. but it not possible to measure porosity with the non-sintered samples since they can not be cut/sectioned to give the necessary size profile.
- It is stated that the thickness of the final samples, both inkjet and bar-coated, was approximately 100 µm. However, the thickness for inkjet printing seen from Fig. 4c is about 600 μm,please check the scale bar.
corrected
- What is the relation between the high electrical resistivity and porous density/pore sizes? Also, what is the purpose of fabricating copper samples with high electrical resistivity?
We have given pore size data, roughness etc in Fig 5 and resistance in Fig 6. It is clear that it plateaus. The purpose to was to show how low laser power was capable of sintering multiple layers. It is not an optimisation study. This paper does show, however, that copper can be sintered in many layers and maintain structure and conductivity without the need for inert atmosphere sintering.
- The texts in Fig.5 are hard to been seen clearly.
Corrected
- In the conclusion, it is stated that “A novel low-power (1 W) sintering process has been successfully demonstrated…”. Whereas, the laser power used in this study is 2.5 W.
corrected
- Please complete information of author contributions.
Corrected
Reviewer 2 Report
Comments and Suggestions for Authors
Very insightful and useful work in the room temperature AM using 10 W laser for copper materials. Few minor concerns should be addressed before publication:
1. Please include copper 3D printing via the electrochemical method in the introduction as another reference. Laser sintering in this paper expands possibilities beyond what was done with the prior approach.
2. What is the resolution of 3D-printed features? Only bar shapes are shown. Can you print any shape?
3. What is the mechanical strength? Please present Tensile testing or at least discuss what is expected under mechanical loading.
4. What is the rationale for using a 10W laser? What will happen if the wattage is reduced or increased slightly?
5. How can you optimize the experimental parameters to yield the best results?
Author Response
REVIEWER 2
Very insightful and useful work in the room temperature AM using 10 W laser for copper materials. Few minor concerns should be addressed before publication:
- Please include copper 3D printing via the electrochemical method in the introduction as another reference. Laser sintering in this paper expands possibilities beyond what was done with the prior approach.
There are 11 references that also include this technique but it is now listed in the first paragraph for completeness.
- What is the resolution of 3D-printed features? Only bar shapes are shown. Can you print any shape?
Not yet. We have focussed on simple layered structures to allow inkjet to be compared directly to bar coating but, in principle, the inkjet could incorporate 3D shapes.
- What is the mechanical strength? Please present Tensile testing or at least discuss what is expected under mechanical loading.
We don’t have this data yet. We did, however include indentation, surface roughness and creep distance as measures of structural performance.
- What is the rationale for using a 10W laser? What will happen if the wattage is reduced or increased slightly?
We have discussed how power vs speed etc impact track sintering in Fig 7/8. The rationale is clearly how to reduce power as low as possible, to avoid ablation etc whilst also still able to create a conductive track.
- How can you optimize the experimental parameters to yield the best results?
This is a first paper exploring how a low power laser might be able to create conductive multilayered structures. It is far from optimised but the novelty lies in the demonstration that it is possible, it doesn’t require inert sintering environments and only requires low powers (to avoid melting multiple layers).
Round 2
Reviewer 1 Report
Comments and Suggestions for Authors
Necessary modifications were completed.
Author Response
We're glad the reviewer agrees with the changes.